1 2

3 4

10

11

12 13

## HAI –

A new, airborne, absolute, twin dual-channel, multi-phase TDLAS-hygrometer

Bernhard Buchholz<sup>1, 2, 5</sup>, Armin Afchine<sup>4</sup>, Alexander Klein<sup>1</sup>, Cornelius Schiller<sup>1</sup><sup>4</sup>, Martina Krämer<sup>4</sup>, Volker Ebert<sup>1,2, 3</sup> <sup>1</sup> Physikalisch-Technische Bundesanstalt Braunschweig, Germany <sup>2</sup> Physikalisch Chemisches Institut, Universität Heidelberg, Germany <sup>3</sup> Center of Smart Interfaces, Technische Universität Darmstadt, Germany <sup>4</sup> Forschungszentrum Jülich, IEK-7, Germany <sup>5</sup>currently at Department of Civil and Environmental Engineering, Princeton University, USA. tdeceased

Corresponding author: volker.ebert@ptb.de

# 14 Abstract

15 The novel Hygrometer for Atmospheric Investigations (HAI) realizes a unique concept for simultaneous 16 gas-phase and total (gas-phase + evaporated cloud particles) water measurement. It has been developed 17 and successfully employed for the first time on the German HALO research aircraft. This new instrument 18 combines direct Tunable Diode Laser Absorption Spectroscopy (dTDLAS) with a first-principle evaluation 19 method to allow absolute water vapor measurements without any initial or repetitive sensor calibration 20 using a reference gas or a reference humidity generator. HAI contains two completely independent dual-21 channel (closed-path, open-path) spectrometers, one at 1.4 µm and one at 2.6 µm, which allow together to 22 cover the entire atmospheric H<sub>2</sub>O range from 1 to 40 000 ppmv with a single instrument. Both spectrometers 23 comprise each a separate, wavelength-individual extractive, closed-path cell for total water (ice and gas-24 phase) measurements. Additionally, both spectrometers couple light into a common, open-path-cell outside of the aircraft fuselage for a direct, sampling-free and contactless determination of the gas-phase water con-25 26 tent. This novel twin dual-channel setup allows for the first time multiple self-validation functions i.e. in 27 particular a reliable, direct, in-flight validation of the open-path channels. During the first field campaigns, 28 the in-flight deviations between the independent and calibration-free channels (i.e. closed-path to closed-29 path and open-path to closed-path) were on average in the 2% range. Further, the fully autonomous HAI 30 hygrometer allows measurements up to 240 Hz with a minimal integration time of 1.4 ms. The best preci-31 sion is achieved by the 1.4 µm closed-path cell at 3.8 Hz (0.18 ppmv) and by the 2.6 µm closed-path cell at 32 13 Hz (0.055 ppmv). The requirements, design, operation principle and in-flight performance of the hy-33 grometer are described in this work.

## 34 1. Introduction

Water vapor is in many ways one of the most important measurand for atmospheric investigations (Ludlam, 1980; Möller et al., 2011; Ravishankara, 2012). Water is the most important greenhouse gas (Kiehl and Trenberth, 1997), and known as a key atmospheric coupling element of almost all microscopic (e.g. droplets/ice crystals formation), macroscopic (e.g. clouds/precipitation) and global processes (e.g. hydrological cycle). Therefore, it is strongly related to the numerous highly relevant topics of atmospheric science and closely related to "climate change" (Held and Soden, 2000; Houghton, 2009; Kiehl and Trenberth, 1997; Maycock et al., 2011). Unsurprisingly, numerous water vapor studies have been carried out targeting its

atmospheric trends and variability (Lu and Takle, 2010; McCarthy et al., 2009; Ross and Elliott, 1996; Scherer 43 et al., 2008; Trenberth et al., 2005; Xie et al., 2011), its influence on transport models (Kiemle et al., 2012; Schäfler et al., 2010), or its impact on radiation balance models (Lockwood, 1990; Ramanathan et al., 1989; 44 45 Schneider, 1972). One reason for the high complexity of atmospheric water vapor is that it is one of the few 46 atmospheric molecules that appears in all three phases. Water in the gas-phase is a very strong infrared 47 absorber and significantly impacts atmospheric energy fluxes through latent heat transfers by the different 48 phase transitions. Condensation to the liquid phase or freezing to solid particles leads to effective scattering 49 of solar radiation, which directly raises links to the formation process of cirrus clouds (Krämer et al., 2009; 50 Spichtinger et al., 2004). These relationships show the complexity from a theoretical as well as a modeling 51 point of view. Today, however, the quality e.g. particularly accuracy and comparability of atmospheric 52 water measurements frequently limit a better understanding of key atmospheric processes (Krämer et al., 53 2009; Peter et al., 2006; Scherer et al., 2008; Sherwood et al., 2014). Despite the outlined importance and the 54 large effort invested in the developments of hygrometers in recent years, water vapor remains a target mol-55 ecule that is very difficult to measure accurately. 56 Several major issues exacerbate water vapor measurements. Atmospheric water vapor encompasses a very 57 large concentration range: 3 - 40 000 ppmv from troposphere up to the lower stratosphere. The spatial fluc-58 tuations of H2O in the atmosphere are high, which leads on fast aircraft (approx. 800 km/s cruising speed) to

- highly dynamic H2O variations of up to several 1000 ppmv/s in the gas-phase and up to several 60 10 000 ppmv/s for total water measurements. These issues require, especially for aircraft based hygrome-61 ters, very high time resolution in combination with very high precision and accuracy. Additionally, since 62 water vapor readily changes from one phase to the other, it would be extremely helpful if hygrometer were 63 able to differentiate between the phases or could differentiate at least between water vapor and total water 64 in order to minimize systematic uncertainties caused by the sampling process. Last but not least, water vapor is very effectively absorbed from nearly any surface. This challenges in a highly complex manner not 65 66 only the entire gas sampling system but also the calibration infrastructure which is typically required for most hygrometers. By waiving the entire calibration process, special hygrometers (Wolfrum et al., 2011) 67
- circumvent all calibration related issues efficiently which will explained in chapter 3.2.

This brief compilation illustrates the complex challenges associated with developing a water vapor instru-70 ment, especially if it should be able to measure in tropospheric and stratospheric atmospheric conditions. 71 Numerous (mostly single-channel) hygrometers have been developed in the last decades with various ad-72 vantages and drawbacks (see (Wiederhold, 1997) and e.g. (Buck, 1985; Busen and Buck, 1995; Cerni, 1994; 73 Desjardins et al., 1989; Diskin et al., 2002; Durry et al., 2008; Ebert et al., 2000; Gurlit et al., 2005; Hansford 74 et al., 2006; Helten et al., 1998; Hunsmann et al., 2008; Karpechko et al., 2014; Kley and Stone, 1978; May, 75 1998; Meyer et al., 2015; Ohtaki and Matsui, 1982; Roths and Busen, 1996; Salasmaa and Kostamo, 1986; 76 Schiff et al., 1994; Silver and Hovde, 1994b, 1994a; Thornberry et al., 2014; Webster et al., 2004; Zöger et al., 77 1999a, 1999b)). Consequently, the question should be raised from the opposite point of view: What are the 78 important and required properties to be covered and combined for a near-universal "Hygrometer for At-79 mospheric Investigation" to serve as an innovative and cutting-edge tool to explore open and new scientific

questions related to atmospheric water vapor?

## 2. <u>Requirements for HAI, a "Hygrometer for Atmospheric Investigation"</u>

## 83 2.1. Specific instrumental boundary conditions defined by the atmosphere

84 Currently, atmospheric water vapor data are generated from two instrument types: Remote sensing ap-85 proaches (like satellites (Oelhaf et al., 2004) or ground-based - often FTIR based - monitoring stations 86 (Zachariassen et al., 2003)) or in-situ hygrometers deployed directly inside of the environment to be meas-87 ured. The latter are roughly separated in ground-base "weather stations", balloon-borne radiosondes (Miloshevich et al., 2006) and hygrometers on airborne carriers like aircraft (Marenco et al., 1998). While 88 89 ground-based stations permanently provide local information, radiosondes deliver "only" one point like 90 vertical profile per ascend/ descend but for heights up to 50 km above ground. Airborne vehicles like air-91 craft and helicopters, however, combine the possibility for arbitrary flight paths and localized measure-92 ments, e.g. for cloud investigations, with a beneficial long operation range. In particular the most modern 93 research aircrafts such as the German HALO aircraft (Anon, 2014; Krautstrunk and Giez, 2012) or the US 94 American HIAPER (Anon, n.d.) bridge the gap and combine broad spatial coverage (> 10 000 km), high 95 altitudes (up to 15 km), and large payloads of up to 3000 kg payload, in a favorable pressurized, and airconditioned cabin. The relatively high traveling speed of these aircraft (of up to 230 m/sec), however, also 96 97 causes disadvantageous influences on the measurements themselves, which are difficult to take care of 98 compared to the very slowly moving (low m/sec range), quasi-static balloon-borne radiosondes. Even rela-99 tively simple meteorological variables such as air temperature and pressure need complex retrieval algo-100 rithms (Giez, 2012) if measured on a high speed aircraft. In short, aircraft in the contemporary working 101 equipment are indispensable for both the investigation of spatially confined effects too small to retrieve 102 from remote sensing data and the validation of the remote sensing instruments (Oelhaf et al., 2004). It is 103 highly desirable for an airborne system to be constructed in a way that it operates on the ground in the 104 same manner as in flight and that every change of environment and "boundary conditions" of the instru-105 ment is logged. This allows, besides multipliable deployments e.g. in ground based stations, extensive vali-106 dation as well as laboratory comparisons with other instruments and avoids systematic, barely detectable 107 deviations only occurring in flight. This notion related to water vapor leads directly into the everlasting discussion about sampling via open-path systems operation in the free flow versus closed-path systems 108 109 extracting the air to be analyzed into the instrument. Open-path hygrometers offer numerous great benefits such as: the prevention of any sampling errors or uncertainties (caused by surface absorbing effects) as well 110 111 as the high response time that is limited only by the transfer functions of optical or electrical components 112 but not by the gas exchange rate. The latter circumvents the complicated and adulterant deconvolution of 113 smoothing effects with time-response functions caused by a sampling system. On the other hand, the 114 boundary conditions such as gas pressure and gas temperature as well as possible spatial inhomogeneities 115 of both parameters are difficult to accurately take into account for an open-path system. Additionally, an 116 airborne open-path sensor has to operate in harsh boundary conditions, i.e. over a large range of tempera-

117 tures (-80 to 50 °C), pressures (70 to 1000 hPa), for large ram pressures (900 km/h gas velocity) and mechan-

- ical stress through accretion of ice or liquid water.
- The major problem of all present open-path systems is their highly complex calibration, or even just valida-
- tion, since realistic flight conditions, in particular the dynamics, are extremely difficult to realize in the lab
- with sufficient accuracy. A direct metrological link to test the inflight performance, e.g. a dynamic calibra-
- tion facility for open-path hygrometers, is therefore missing.
- Closed-path systems, on the other side, are simply installed inside an air-conditioned cabin, in a much more 124 protected environment. Gas is sampled with a suitable inlet and led via a tubing system to an "internal" 125 measurement chamber, such as an optical absorption cell or a suitable "cavity", e.g. for a dew point mirror hygrometer. On one hand, it is much easier to accurately control and maintain the physical boundary con-126 127 ditions of the sample gas, e.g. temperature, pressure, flow, etc. in the measurement volume. On the other 128 hand, it is difficult to ensure and maintain a representative sampling process and to quantify and correct 129 sampling related deviations. These maybe caused by adsorption and desorption effects, which occur on all 130 surfaces of the sampling system, and have to be carefully minimized by heated (HAI ≈80 °C) stainless steel 131 sampling pipes, along with an instrument design ensuring high gas flows (HAI ≈100 liter/min) under all 132 flight conditions.
- Gas-phase H2O measurements in clouds are often carried out via backward facing sampling inlets. But such 134 inlets are readily sampling small liquid or solid water particles possibly causing systematically positive 135 offsets. For every H2O measurement using airborne extractive (=closed-path) instruments on aircrafts, it needs to be taken into account that the instruments response reflects contributions from the sensor element 136 137 itself as well as the sampling/tubing system and their dynamic properties. One indisputable, major ad-138 vantage of typical extractive instruments is the possibility for careful tests outside of the aircraft, i.e. in a 139 hangar or laboratory. But, to take full advantage of this, it is desirable for a sophisticated instrument to integrate supervising and monitoring functions in a way that a performance comparable to the laboratory can 140 141 be ensured during any inflight situation. This generates the great benefit of transferring the performance 142 from the laboratory to the field situation, quite similar how a metrological transfer standard is typically 143 used. This directly reinforces the question, how to assess the accuracy of airborne hygrometers.

## 144 **2.2.** Accuracy and state of the art instrumentation

In general, the highest measurement and preparation accuracy is realized by the validated primary stand-146 ards of national metrology institutes such as PTB (Germany) or NIST (USA). The international, metrological 147 water vapor scale is defined by traceable primary water vapor generators (Brewer et al., 2011). The mixing 148 ratio range, required to cover the entire tropospheric and stratospheric coverage of about 3 - 40000 ppmv, is 149 realized by a combination of generators based on different physical principles. Their typical uncertainty is 150 in the order of 0.5% relative, but varies with the physical principle used and their realization (Brewer et al., 151 2011; Buchholz et al., 2014a; Mackrodt, 2012). In other words, it is not possible to validate any hygrometer 152 with better accuracy due to the lack of a suitable accurate reference.

Comparing the available metrological accuracy to some results from field comparisons of airborne hygrom-154 eters demonstrates the large potential for improvement. For example, long-term (> 10 years) change studies 155 of stratospheric H2O (Oltmans and Hofmann, 1995; Rosenlof et al., 2001; Solomon et al., 2010) suffer from 156 significant, difficult to quantify relative deviations between different instruments in the range of 50-100% 157 (Fahey et al., 2014; Peter et al., 2006; Vömel et al., 2007) which recent studies such as (Rollins et al., 2014) 158 confirm ( $\pm 40-50\%$  for < 3ppmv and  $\pm 20\%$  for > 3ppmv). Radiosonde comparisons with polymer sensors and 159 chilled dew point mirror hygrometer, covering the entire troposphere and lower stratosphere region such 160 as (Miloshevich et al., 2006), show averaged overall agreement in the 10% range (but also, local deviations 161 in the 30% range). These deviations are quite common for many airborne campaign results e.g. (Smit et al., 162 2014) and become even worse when focusing on the relative deviations in regions containing highly varia-163 ble H2O structures. Hence in 2007, an international comparison exercise, "AquaVIT" (Fahey and Gao, 2009), was organized to compare the world's best airborne hygrometers under well-controlled, quasi-static, 164 165 equivalent conditions to evaluate the accuracy under well controlled laboratory conditions, without the 166 influence of any typical airborne sampling and dynamic effects. AquaVIT comprised 22 hygrometers (tuna-167 ble diode laser spectrometers, TDL, dew or frost point mirror hygrometers, D/FPH, Lyman alpha fluores-168 cence and absorption hygrometers, LAFH and other principles) from 17 international research groups. The 169 instruments were categorized in well validated "core" instruments (APicT, FISH, FLASH, HWV, JLH, CFH, 170 see (Fahey and Gao, 2009) for details) and "younger, less mature" non-core instruments. Even the core-171 hygrometers deviated in the important 1 to 150 ppmv WVMR range by up to  $\pm$  10 % from the mean. In oth-172 er words, core instruments differed by up to 20% from each other, even under quasi-static conditions. Other, less representative and extensive comparisons such as (Hoff, 2009; Mangold and Wodca Team, 2003) 173 174 yielded similar results.

175 Of course, the assessment of the required accuracy depends strongly on the purpose of the data. In terms of 176 climatologies and strongly averaged or coarse validation studies such deviations can be acceptable, but as 177 far as e.g. retrieval models for satellite data improve, such uncertainties and deviations can become critical. 178 In many other cases such as the currently often discussed atmospheric super saturations (Peter et al., 2006), 179 the instrument uncertainties prevent deeper investigations and therefore a better understanding. Reconsid-180 ering the entire situation and perceiving that after so much development effort over the past decades these 181 deviations remain quite high, leads to the inevitable question of the concealed, common impact factors. 182 Contemplating the typical metrological efforts needed at national metrology institutes (NMI) to generate an 183 accurately humidified gas stream (with a sub-percent uncertainty) suggests that the uncertainties generated 184 by typical calibration processes under field conditions could be a major contribution to these hygrometers 185 deviations found in AquaVIT and other studies. In particular, comparing the performance and strategies of 186 lab-based, metrological and portable field calibration facilities (Friehe et al., 1986; Helten et al., 1998; 187 Podolske et al., 2003; Smit et al., 2000; Smorgon et al., 2014; Zöger et al., 1999b) show three significant dis-188 crepancies: required time for calibration, frequency of calibration, and traceability of the humidity reference 189 itself. Calibrations in the low concentration ranges at NMIs take several hours up to days per individual 190 humidity value. During airborne campaigns, however, calibrations often have to be realized (for practical 191 reasons) in a short time, certainly less than a few hours (max) for a large number of concentration steps Page 5

192 often including several pressure levels, thereby taking the obvious risk that the instrument/reference is not 193 fully stabilized or equilibrated. Ideally, the time between two calibrations should be shorter than the ex-194 pected time required for a drift/change exceeding the boundaries of the instrument uncertainties. Some 195 airborne instruments require for the same reason calibrations before and after each flight in order to inter-196 polate between both calibrations (Zöger et al., 1999b). Some even work with in-flight calibrations 197 (Kaufmann et al., 2016) sacrificing measurement time and shifting the accuracy issue to the necessary air-198 borne H2O-source. Undoubtedly, many of these instruments have benefits e.g. in terms of precision, space, 199 weight, prime cost etc., which justifies the calibration effort. However, vice versa, it is condoned that the 200 calibration process is hampered and turns out to be the major influence on the accuracy of such a sensor.

Lastly, it seems necessary to implement a traceable link to the metrological humidity scales to improve the overall accuracy of airborne hygrometry (Joint Committee for Guides in Metrology (JCGM), 2009). By realizing an unbroken chain of calibrations, it is possible to link the instrument performance and the metrological water scale respectively the SI system of units. This also guarantees an accurate measurement/generation value with defined uncertainties.

To summarize: Fulfilling all these demands in the field similar like in a NMI laboratory is a tough task. However, as discussed later, many of the covered issues can be circumvented using first principle techniques like dTDLAS (Ebert and Wolfrum, 1994; Schulz et al., 2007) to realize optical, absolute hygrometers which avoid over certain defined operating ranges any water vapor sensor calibration.

210 From a user's point of view, precision and response time of an airborne hygrometer appears equivalent to accuracy if he is interested in fine structure resolving data. Precision and response time are under certain 211 212 circumstances reciprocally correlated to each other (Allan, 1966). Typical figures for response time of air-213 borne hygrometer in the literature are 0.5-1 Hz (Petersen et al., 2010; Szakáll et al., 2004; Zöger et al., 1999b); some instruments deliver faster data 4 Hz (Weinstock et al., 1994) or special instruments up to 25 Hz 214 215 (Zondlo et al., 2010). Typical precisions at 1 Hz are in 0.1-0.2 ppmv range (Sargent et al., 2013; Zöger et al., 216 1999b; Zondlo et al., 2010). While in the stratosphere (<10 ppm), the precision certainly can become a limit-217 ing factor. This is much less the case inside clouds or within the troposphere, where frequent, very strong spatial variations (up to 1000 ppmv per 100 m flightpath in the gas-phase, or up to 20000 ppmv per 100 m 218 219 during total H2O phase) pose a larger problem. An instrument with a time response of just a few Hz causes 220 significant under-sampling which can lead to strong aliasing effects at the high velocities (approx. 700-221 900 km/h) of many research aircrafts. Important under such conditions is the instrument's linearity to accu-222 rately cover the entire H<sub>2</sub>O concentration range up to five magnitudes for total water vapor measurements.

Numerous additional requirements have to be fulfilled by an airborne instrument to ensure a successful
operation. Due to the high operation costs for aircrafts and the high scientific demand, H<sub>2</sub>O data have to be

225 measured continuously without any interruptions. The instrument thus has to be highly reliable, robust,

and require low maintenance. The restrictions in weight and space as associated with operation on an air-

227 craft result in the necessity for the device to be of a compact and lightweight construction. The utterly com-

- plex and mandatory certification process (at least in Germany) quickly enforces an instrument design freeze
- before a campaign; this results in very stiff constraints for improvements/repairs during a campaign.

## 231 3. Design decisions and approach

HAI reflects on these by providing four independent but coupled spectrometers in one single housing and
by simultaneously combining open- with closed-path measurements in one single instrument. For all
channels the evaluation is done with one common spectroscopic method: calibration-free direct Tunable
Diode Laser Absorption Spectroscopy (cal-free dTDLAS).

### 3.1. Direct Tunable diode laser absorption spectroscopy (dTDLAS)

The requirements for fast measurements and high chemical selectivity in combination with a robust and small system calls to choose a contact-less, spectroscopic (hence optical) measurement technique rather than 239 240 contact -mediated sensing methods such as dew point mirror hygrometer (DPH) or capacitive polymer 241 sensors e.g. "Humicap" (Salasmaa and Kostamo, 1986; Smit et al., 2000). These are quite often used in air-242 borne hygrometry and especially in meteorological environments (Anon, 2010; Busen and Buck, 1995; 243 Hansford et al., 2006; Wiederhold, 1997). Optical hygrometers can be set up to become quite immune to 244 hydrophilic/hydrophobicsubstances (unavoidable in the vicinity of aircraft) as well as particles (dust, soot, ice, etc.) contained by the gas to be analyzed. These capabilities were e.g. extensively demonstrated via 245 measurements inside of combustion processes in industrial power plants (Ebert et al., 2000a; Schlosser et al., 246 247 2002; Sun et al., 2013; Teichert et al., 2003). Even response times in the several 10 kHz range could be 248 demonstrated recently by (Witzel et al., 2013) for measurements in combustion engines.

Tunable diode laser absorption spectroscopy (TDLAS) is a powerful as well as versatile diagnostic technique which is frequently employed in the near infrared spectral range and led to numerous applications in atmospheric hygrometry (Diskin et al., 2002; Fahey and Gao, 2009; Gurlit et al., 2005; May, 1998; Schiff et al., 1994; Thornberry et al., 2014) . Advantageous diode lasers properties are the very high spectral resolution and power density, the continuous tuneability in combination with interesting technical features such as low cost, very low size/weight/power consumption, long life time, excellent beam quality, and optical fiber coupling to name just a few.

The typical setup and working principle of a TDLAS instrument has been frequently described in detail 257 (Lackner, 2011; Schiff et al., 1994; Schulz et al., 2007; Werle, 1998). Therefore, only the HAI design relevant 258 topics are discussed here. Important for an understanding of the novel HAI instrument is the classification 259 of TDLAS instruments by their optical detection schemes in classical single- (Ebert et al., 2000b) or multi-260 path (Gurlit et al., 2005; Hunsmann et al., 2008; Lübken et al., 1999; May, 1998; McManus et al., 1995) beam setups- Further major categorizing distinguishes between the wavelength-modulation schemes like single 261 262 modulation frequency = direct TDLAS or dTDLAS (Ebert and Wolfrum, 1994) or double modulation schemes like wavelength modulation spectroscopy WMS (Podolske and Loewenstein, 1993; Silver, 1992; 263 Silver and Hovde, 1994a; Silver and Zondlo, 2006; Vance et al., 2011; Webster et al., 2004). WMS which is 264 265 often used for very compact airborne sensors, provides on the first glance higher sensitivities by using lockin technologies to efficiently filter noise. This, however, sacrifices the possibility of direct physics-based 266 267 quality and reliability checks, since the actual measured WMS raw signal contains less spectral information Page 7

than a dTDLAS raw signal. This aggravates or sometimes even prevents detailed signal analysis based on fundamental physical explanations. Using dTDLAS instead with a special, but less common, first principles evaluation procedure (Ebert and Wolfrum, 2000; Farooq et al., 2008; Mihalcea et al., 1997; Schulz et al., 2007) yields a sophisticated evaluation, characterization, and quality management and a holistic view on the physical principles behind the data. This circumstance can even be used to avoid typical calibration procedures with reference gas standards.

## 275 Explanation of the term "calibration-free"

The term "calibration-free" is often used in different communities with dissimilar meanings. To distinguish 276 one should consider how calibration is defined by metrology (JCGM 2008, 2008): "calibration (...) in a first 277 278 step, establishes a relation between the measured values of a quantity with measurement uncertainties pro-279 vided by measurement standards (...), in a second step, uses this information to establish a relation for ob-280 taining a measurement result from an indication (of the device to be calibrated)". In other words, an in-281 strument with a deterministic relation between indication and quantity can be calibrated, without knowing 282 the physics behind it. This particularly allows for compensating of non-linearity, offsets, drifts or response 283 changes over time as long as they are stable, predictable, or can be extrapolated. We use the term "calibra-284 tion-free" to emphasize that HAI does not rely on such a correction process. The hygrometer described in 285 this paper has never been calibrated or adjusted to a water vapor primary standard. Of course, parameters 286 like gas pressure and temperature that are used for the calculation of the water vapor content via a first principle model are measured with calibrated sensors. This is done from a practical point because a) prima-287 288 ry standards for temperature and pressure are by themselves large facilities and b) the influence in the final 289 uncertainty budget doesn't justify that approach. Calibration-free doesn't mean that the whole setup only uses primary principles. The whole idea behind traceability (JCGM 2008, 2008) is to use other sizes, higher 290 291 in the hierarchy of the SI units of course, to generate/analyze the target value. Metrologists name that an 292 unbroken chain of measurements. To visualize, this means that the first water vapor value delivered by the 293 HAI instrument is the final value. Everything has to be characterized in advance on such a level that the 294 first measurement value is determined within its uncertainty limits. Therefore, since there was no calibra-295 tion, it's termed calibration-free. 296 Obviously, the calibration-free approach can always be enhanced with a calibration (such as (Muecke et al.,

1994)) at any time (Buchholz et al., 2013b), even after a campaign if this seems advantageous, since the requirements for a calibrated instrument are lower than for a calibration-free instrument.

### 299 3.2. Non-calibrated direct TDLAS (dTDLAS)

The principle of non-calibrated absolute dTDLAS is very briefly presented in the following section. More detailed information regarding TDLAS is referred to in the above-mentioned literature. The sketch in Figure 1 shows the schematics of a dTDLAS spectrometer with two independent channels. For low light intensities  $I_0(\lambda)$  in the mW-range, the transmitted light  $I(\lambda)$  can be described by the extended Lambert-Beer equa-

- tion (Equation 1) including possible disturbances or the absorption measurement by background radiation
- E(t) or broadband transmission losses Tr(t).
- Equation 1:  $I(\lambda) = E(t) + I_0(\lambda) \cdot Tr(t) \cdot exp[-S(T) \cdot g(\lambda \lambda_0) \cdot N \cdot L]$
- By applying the ideal gas law Equation 1 can be used to retrieve the H<sub>2</sub>O volume mixing ratio c.
- Equation 2:  $c = -\frac{k_B \cdot T}{S(T) \cdot L \cdot p} \int ln \left( \frac{I(v) E(t)}{I_0(v) \cdot Tr(t)} \right) \frac{dv}{dt} dt$

The amount fraction c is in metrological units officially specified as [mol/mol = mol absorber per mol gas] 310 which is in the environmental community better known as "volume fraction" e.g. in units of ppmv or 311 Vol.-%. The term  $\frac{dv}{dt}$  is called the *dynamic tuning coefficient* of the used laser. It can be determined experimen-312 tally and is directly linked to the SI units (length) by using the Airy signal of the laser light passing through a planar, air-spaced etalon (Ebert and Wolfrum, 2000; Schlosser et al., 2002). Yet, unpublished, on-going, 313 and long-term <sup>av</sup>/<sub>4</sub> measurements for the used 1.37 µm DFB laser type over several years indicate a long-term 314 315 stability of its tuning characteristics better than 1%, which is within the current uncertainties of the tuning characterisation.  $k_B$  is the Boltzmann constant and L is the optical path length. S(T) is the line strength of the 316 317 selected molecular transition (see chapter below) and therefore a physical property of the molecule to be 318 measured. The gas pressure (p) and gas temperature (T) can be accurately acquired in a closed-path cell; the quality of the respective measurements in the open-path cell are discussed in the following chapter describ-319 320 ing the construction of the open-path sensor. Equation 2 also "explains" the term calibration-free quantita-321 tively as there are no other "hidden" parameters used to derive the water vapour concentration which re-322 quire a calibration.

### 323 **3.3.** Absorption line selection

Suitable absorption lines have to be selected for a specific application by several criteria (Wunderle et al., 325 2006) (Wagner et al., 2012). Besides a line strength maximization to ensure high sensitivity, other important 326 parameters have to be taken into account. For atmospheric measurements, the cross sensitivity to other 327 gases such as CO2 needs to be minimized. This ensures a better control of the fitting process due to the few-328 er degrees of freedom. Similarly, the line should be isolated from other lines to simplify the retrieval of the 329 baseline function. For the open-path measurements, it is highly important to minimize temperature de-330 pendence of the line strength in order to minimize the influence of gas temperature uncertainties. Lastly, 331 sometimes the primary constraint is the availability of suitable laser diodes and additional accessories such 332 as fibers and optic components. As the certification for airborne instruments nearly prevents improve-333 ments/repairs during a campaign, all components need to be very reliable. For HAI, we selected two specif-334 ic water lines at 2596 nm and 1370 nm in two wavelength ranges, which also had been used before 335 (Buchholz et al., 2012, 2013b; Ebert, 2006; Ebert et al., 2004; Fahey and Gao, 2009; Hovde et al., 2001; Hunsmann et al., 2008; May, 1998; Seidel et al., 2012; Witzel et al., 2012; Wunderle et al., 2008) and for which 336 337 improved spectral parameters were generated (Hunsmann et al., 2006). The 2.6 µm laser is not fibercoupled, doesn't have an optical isolator, is less stable in terms of temperature fluctuations and has a lower 338 Page 9