# Peer review of "A new, airborne, absolute, twin dual-channel, multi-phase TDLAS-hygrometer"

_Atmospheric Measurement Techniques, 2016_

## Referee Comment (RC1) · Anonymous Referee #1 · 28 Jun 2016

This manuscript describes a multi-channel spectroscopic measurement system for the airborne detection of water vapor and total water content. The authors describe in detail the requirements for such a system, the setup characterization of their system, and example data from flight testing of their system. The manuscript has strong technical merit and represents a significant contribution to scientific progress. However, the manuscript suffers from some balance issues with respect to material, and the writing has some fluency issues. While I strongly believe this manuscript is ultimately worthy of publication, I recommend that the manuscript be returned to the authors and reconsidered after the address of some of the below issues.

Major comments:

- The authors spend a proportionally large amount of the manuscript on what I

would consider background. Section 2 seems to act as a second, instrument-centric introduction. While this background is generally accurate (save some specific comments below), it seems more fitting for the start of a review article, which does not seem to me to be the intent of this manuscript. I would recommend greatly condensing section 2 and combining it with section 1 to create a better proportioned background section.

- I would shift the extra room in the manuscript to adding additional details with respect to the instrument system itself, which I found to be somewhat sparse at points. In Sect. 4.1, the authors assert "Many of the modules contain innovative developments, which are or will be published individually to prevent an overload of technical details in this paper." I believe that this argument is better suited for analysis-centric papers, whereas being a technical manuscript in AMT, this is exactly what this paper should discuss

- There are numerous places (I tried to list them in the comments below) where assertions were made that were not fully explained or where additional evidence would better support the assertions. For example, the authors make a strong argument for the "calibration-free" method for their instrumentation. While I agree there is sound theoretical evidence for this approach, I feel it would make a stronger argument to have, not a calibration, but a validation for the approach other than the internal validation which is susceptible to common biases. This also seems to be easy to accomplish for the closed-path systems. Other examples are listed in the specific comments.

- The manuscript also needs additional editorial support before it will be ready for publication. While most of the paper is intelligible, there are many instances of awkward language and some instances where this inhibits comprehension. I have denoted some cases below, but it is not an exhaustive list.

Specific comments:

- Line 51: missing parens

- Line 58: 800 km/hr?

- Line 93-94: cite aircraft by organization (and perhaps in addition nationality) for clarity

- Line 94: It seems there should be a better HIAPER reference

- Line 100-101: meaning of "aircraft in the contemporary working equipment" unclear, translation issue? Could not tell from context.

- Line 105: "besides multipliable deployments" confusing word choice

- Line 196-197: Cite Thornberry et al. (2013) or Thornberry et al. (2014) for in flight cal system

- Line 206: NMI?

- Line 350's: The authors should provide additional information about the lasers (e.g. manufacturer, power) and the optics (e.g. focal lengths).

- Line 390: does 10-4 refer to optical depth?

- Line 405: Relative internal vs external path lengths for the 2.6 $\mu$m laser? Only the 1.4 $\mu$m laser is specifically mentioned here. Is the 2.6 $\mu$m optics the same?

- Line 422: Cite 2014 reference for optical airborne pressure measurements here

- Line 487: This doesn't make sense to me. Where is the 70 ms coming from? If only 1/6 of the spectrum is necessary, why scan more than that?

- Line 501-506: Why does WMS specifically require lower fringe levels? This requirement would involve many factors, and the authors do not present any evidence or reference for the claim.

- Line 509: ramp -> ram?

- Line 511-512: OP 6x greater 1s residual that CP? This doesn't make sense with the Sect. 5.2 results

- Line 512-513: How does the wind speed affect this? Is it pressure deviation? Cavitation causing misalignment? Wall effects from the airfoils? Have you determined this from performing speed experiments on the aircraft?

- Line 513-520: This seems an odd place for this discussion. Also, much of the temperature and pressure difficulty with open path systems arises from the uncertainties in the temperature and pressure fields themselves, not instrumental artifacts, which would result in the same issue here. Spectral effect in temperature and pressure can be minimized through spectral line choice, which would also be similar for either method.

- Line 543-546: This precision is for 1.4 $\mu$m or 2.6 $\mu$m system? What is the precision of the other? Maybe a table to summarize which precisions/detection limits correspond to which channel

- Line 598-601: it would be nice to see a figure with a greater summary of the intercomparisons between the 4 channels than the singular example shown in figure 10

- Line 604-609: this statement should be made with the caveat that the 20

- Line 637-652

  – The instrument in (c) has flown many campaigns on the NASA ER-2 (see JPL Laser Hygrometer, or JLH).

  – The authors omit the NASA DLH hygrometer (Diskin et al., 2002), which has frequently flown missions for over 20 years.

- Line 644-645: again recommend that aircraft be referred by their affiliation, not nationality. HAIPER -> HIAPER.
  - This 2

- Line 678-684: this would benefit from an additional figure

- Figure 1: I'm not sure I see the need for this figure, other than to show that the same laser can be used for multiple channels simultaneously. I recommend removing it. Maybe the intent of this figure could be added to Fig. 4 by adding the parasitic water and stabilization channels.

- Figure 2: This would be more beneficial if this were split into two bottom axes, then zoomed in on each water line, showing only the lines immediately around it.

- Figure 5: I don't feel this figure adds significantly to the manuscript, and I recommend removing it.

- Figure 7: cursing -> cruising

---

## Referee Comment (RC2) · Anonymous Referee #2 · 8 Aug 2016

This paper discusses an aircraft tunable laser spectrometer titled Hygrometer for Atmospheric Investigations (HAI) for in situ water vapor and total water measurements. The paper discusses the four separate measurements that HAI provides via two independent diode lasers (1.4 and 2.6 µm) and two different analysis cells (open and closed path). The authors discuss how HAI will improve upon present field instruments and show some flight data.

Major Issues:

1) The purpose of this manuscript is unclear.

The present paper has 'multi-phase' in its title but does not discuss HAI total water components in sufficient detail. Such details are necessary to understand how the effectiveness of HAI multi-phase measurements. Total water measurements present new challenges such as particle enhancement factors and memory effects that require careful study – and none of that is presented in this manuscript.

Rather, this paper is about how the methodology of multi-channel water measurements provide a more accurate water vapor measurement than prior art. In this light, there is clearer justification for this manuscript – i.e. that measuring water vapor simultaneously via two different wavelengths and two different analysis cells helps constrain measurement uncertainty. However, the authors need to re-write the body of the manuscript to discuss this in more detail. For instance, the Harvard HHH and total water instrument provide a multi-channel measurement but that is not discussed in this manuscript. Also, past aircraft measurements involved multiple instruments on one aircraft – how does HAI do better than these experiments?

2) The term 'calibration-free' is misleading and the authors do not prove HAI is 'calibration-free'.

At the percent or better level (accuracy and/or repeatability), even first-principles, direct absorption measurements need to consider instrumental artifacts such as detector non-linearity and low-pass filtering of the detector signal. Both of these artifacts can be misinterpreted as using the wrong line-shape algorithm. In this way, a first-principles methodology might be more prone to error than one that lumps all translation factors into one multi-parameter function.

It will take some work to prove a system to be 'calibration-free'. Such proof would be measuring a known concentration if non-stick gas (methane for the 1.4 µm channel) at several different concentrations, then repeating this for different temperatures the detector and filtering electronics will encounter.

3) Lines 638-650 have incomplete information and need to be better researched and re-written.

There are at least two other open path aircraft instruments not listed here: DLH (Disken et al.) and HHH (Harvard Anderson Group). Moreover, JLH (part c of author's list) has flown more than 200 times and there is a wealth of information on how it performed against other water instruments and satellite retrievals.

Minor issues:

1) The entire paper needs to be edited to be less conversational. For example, don't use phrases like "Of course" and re-write the sentence in lines 226-228 to be less conversational.

2) Correct spelling of 'Herriott'

3) Define 't' in equation 1.

---

## Referee Comment (RC3) · Anonymous Referee #3 · 8 Sep 2016

Buchholz and co-authors present a thorough and well written description of a thoughtfully-considered 2x2 spectroscopic measurement of water vapor and total water. I recommend publication after they address a few issues detailed below:

My major criticism concerns the total water channels. The inlet design parameters of 80ïĆřC piping temperature and 100 liter/min flow rate (line 130) do not seem appropriate to ensure efficient conversion of condensed water to vapor prior to quantification in the closed-path cells. Any characterization of condensed phase sampling efficiency or comparison to other measurements during ESMVal would improve confidence in the accuracy of these measurements. If it is known that the total water channels have some inadequacies, this should be explicitly discussed at appropriate points within the manuscript, or alternatively, claims of multi-phase measurement capability should be tempered or removed.

More minor improvements are suggested below.

In the section 'Explanation of the term "calibration-free", the authors are encouraged to add a discussion of the criteria they use to determine when and how frequently they re-characterize the component calibrations within the 'unbroken chain of [metrological] measurements' to assure the continued traceability of direct absorbance measurements.

Reference lists of existing hygrometer publications exclude Zondlo [2010] in line 77 and in the reference cited in lines 94 and 645. Additionally on line 645, the American aircraft model is a Gulfstream-V (500).

In line 650 and following, the May et al. developed hygrometer has flown on many NASA missions, supported by May and Webster as well as by Robert Herman. See, for example, data publication and instrument description found in A Hallar, et al., 2004 JGR-Atmospheres and DE Hagan, et al., 2004 GRL.

In the sentence beginning on line 661, would the .6%/K temperature influence on uncertainty result in a +/- 3% effect on the open-path water vapor measurement (due to the 5K local temperature mesurement uncertainty)?

In line 663, did the authors mean to write Voigt width, rather than Gaussian?

In the paragraph beginning on line 667, the systematic mean difference of the 2.6 closed-path: 1.4 closed path channels is more highly variable during the in-cloud transect and visual inspection would seem to imply that the mean difference undergoes a shift or temporal trend. An effort to filter both data sets to the same sampling intervals and frequencies prior to difference calculation would perhaps allow a more meaningful comparison between the mean channel difference during clear sky and in-cloud operations. It would not be surprising to see that the increased turbulence of in-cloud operations has differential impact on components of the 2x2 hygrometer, increasing the variability and perhaps even the mean of systematic channel differences.

The authors claim in line 714 that the open-path and closed-path measurements show only a 2% difference in the same flight segment, however, Figure 10 shows larger differences, with instantaneous channel differences falling between +/- 3 to 5%.

Among the figures, it would be valuable to add a figure (after Fig. 8?) to show the fit quality of the open-path 2.6 um channel since it is the most likely channel to saturate.

Finally, it would be useful to add a plot of relative channel deviation as a function of water vapor concentration over the same time interval as Figure 10 (13:40-13:55) to show that the channel differences are not related to calibration errors.

---

## Author Comment (AC1) · 5 Oct 2016

We thank all the reviewers for carefully reading our manuscript and for the detailed feedback aimed at helping us to further improve the manuscript. Below we address the raised concerns in a point by point fashion. Changes are highlighted in the attached revised version.

**Anonymous Referee #1**

This manuscript describes a multi-channel spectroscopic measurement system for the airborne detection of water vapor and total water content. The authors describe in detail the requirements for such a system, the setup characterization of their system, and example data from flight testing of their system. The manuscript has strong technical merit and represents a significant contribution to scientific progress. However, the manuscript suffers from some balance issues with respect to material, and the writing has some fluency issues. While I strongly believe this manuscript is ultimately worthy of publication, I recommend that the manuscript be returned to the authors and reconsidered after the address of some of the below issues.

Major comments:

• *The authors spend a proportionally large amount of the manuscript on what I would consider background. Section 2 seems to act as a second, instrument centric introduction. While this background is generally accurate (save some specific comments below), it seems more fitting for the start of a review article, which does not seem to me to be the intent of this manuscript. I would recommend greatly condensing section 2 and combining it with section 1 to create a better proportioned background section. I would shift the extra room in the manuscript to adding additional details with respect to the instrument system itself, which I found to be somewhat sparse at points.*

=> Thank you very much for that comment. Indeed, the background part is extensive - but from our perspective absolutely needed. The airborne water vapor community is quite mature and developed a significant variety of outstanding instruments. But, many instruments are designed for highly specific purposes and thus difficult to compare. We tried to give a good overview on the field in order to better show the novelty of our instrument. Due to the large number of airborne hygrometers it is impossible to mention all instruments – however, our goal was to describe at least the most important representatives for all categories. This brief overview should provide a background for a reader of the current state of development in the airborne hygrometer community and place HAI accordingly in this perspective. This wealth of information might be avoidable for the strongly experienced and specialized readers. But most of the readers don't have this broad background and would welcome the information. Reviewer 2 actually asked – contrary to the first reviewer - for even more information in the introduction. While for you as the first reviewer with obviously a broad knowledge of current activities in the airborne instrument development this information was maybe not necessary, it wasn't enough for reviewer 2 ( see: "Also, past aircraft measurements involved multiple instruments on one aircraft – how does HAI do better than these experiments?"). Therefore, we tried to find the balance between describing background and anticipating enough knowledge about the field. => We revised the introduction.

• **In Sect. 4.1, the authors assert "Many of the modules contain innovative developments, which are or will be published individually to prevent an overload of technical details in this paper." I believe that this argument is better suited for analysis-centric papers, whereas being a technical manuscript in AMT, this is exactly what this paper should discuss.**

=> We agree here in principle. However, some features allowing HAI to measure with an entirely different approach are too complex to all be pressed in a single paper. E.g. the entire procedure how to minimize offsets caused by parasitic effects filled an entire paper (Buchholz and Ebert, 2014b). A fast, 10-line explanation would not justify e.g. why we claim that HAI can calculate (!) its offset uncertainty. This example shows an important implication: Calibrated instruments can easily be calibrated for offsets and therefore an effect like that is typically neglected; however, we showed that these parasitic offsets can cause large offset fluctuations – something which is entirely neglected in the vast majority of the publications we know. Different, but similar to that are gas leakages in spectrometers. We cannot calibrated for that, though, for HAI it was crucial to develop leak-tight, strain-relieved, adjustable fiber-throughput (Buchholz and Ebert, 2014a). It is known that these feed-throughputs are critical. Therefore, a 10-line explanation would not be enough to justify our assumptions (and extensive validations) that our uncertainty budget does not contain a leakage contribution because we can prove (!) that the influence is negligible. We also cannot add in a few lines many other features as complex/advanced as that (e.g. the spectral line stabilization algorithm). Similarly, Rev2 asked for entire temperature/pressure /concentration etc. validations. We don't want this paper to be a collection of "superficial" tests and statements. Instead we see the need to explain the crucial steps to the readers which are needed to demonstrate and justify the performance of HAI and thus need more space for this to explain it in way that it is useful for the reader. HAI should show in every single feature its full performance capability, clearly explained and deep enough that other scientist can learn from the features for their instruments. Our goal is a metrological traceable instrument. But of course this is a 10year long-term goal with several obstacles. If there was a straight way, it wouldn't be clear why the community struggles with large discrepancies shown in AquaVIT-I and similar comparisons.

=> We added more instrumental information where affordable and where the manuscript benefited from them.

**• There are numerous places (I tried to list them in the comments below) where assertions were made that were not fully explained or where additional evidence would better support the assertions. For example, the authors make a strong argument for the "calibration-free" method for their instrumentation. While I agree there is sound theoretical evidence for this approach, I feel it would make a stronger argument to have, not a calibration, but a validation for the approach other than the internal validation which is susceptible to common biases. This also seems to be easy to accomplish for the closed-path systems. Other examples are listed in the specific comments.**

=> Thank you very much for that advice, we revised the text on several positions to clarify.

We agree that a validation is important to prove if a theoretical concept works in reality. One validation, directly relevant for HAI, has already been conducted and published: (Buchholz et al., 2014a). At PTB, a broader metrological validation of this method for much longer time spans is currently ongoing. Eventually, this method validation will benefit HAI. However, since these measurements are not done with each individual channel of HAI but with the single channel instrument named SEALDH-II, we would foresee - if added - distraction of the reader from the goal of that paper which is an overview description of HAI.

**• The manuscript also needs additional editorial support before it will be ready for publication. While most of the paper is intelligible, there are many instances of awkward language and some instances where this inhibits comprehension. I have denoted some cases below, but it is not an exhaustive list.**

=> We have done that in the final version

**Specific comments:**

• **Line 51: missing parens**

=> added

• **Line 58: 800 km/hr?**

=> changed to km/h

• **Line 93-94: cite aircraft by organization (and perhaps in addition nationality) for Clarity**

=> both changed to their official references

• **Line 94: It seems there should be a better HIAPER reference**

=> corrected

• **Line 100-101: meaning of "aircraft in the contemporary working equipment" unclear, translation issue? Could not tell from context.**

=> sentence obsolete due to rewording paragraph

• **Line 105: "besides multipliable deployments" confusing word choice**

=> changed

• **Line 196-197: Cite Thornberry et al. (2013) or Thornberry et al. (2014) for in flight cal system**

=> added

• **Line 206: NMI? => National Metrology Institute**

=> added

• **Line 350's: The authors should provide additional information about the lasers (e.g. manufacturer, power) and the optics (e.g. focal lengths).**

=> Information added

• **Line 390: does 10-4 refer to optical depth?**

=> we added optical density signal-to-noise

• **Line 405: Relative internal vs external path lengths for the 2.6 μm laser? Only the 1.4 μm laser is specifically mentioned here. Is the 2.6 μm optics the same?**

The closed-path cells are similar (same optical path length), while both laser's light-path are coupled in the same open-path White cell.

=> see submitted manuscript line 386, 397,409

• **Line 422: Cite 2014 reference for optical airborne pressure measurements here**

=> done

• **Line 487: This doesn't make sense to me. Where is the 70 ms coming from? If only 1/6 of the spectrum is necessary, why scan more than that?**

=> In this flight situation, only approx. 1/6 of the whole (up- and down-ramp) spectral scan contains the water vapor content information, while the rest is needed for retrieving the baseline, the laser-out region and for situations where gas pressure and water vapor concentrations are much higher (e.g. low flights in warm areas, especially in clouds). This leads to a time resolution (= after averaging) of 70 ms for each reading (1/240 * 1/6 * 50).

=> We rephrased the sentence.

**Why scan more than that?**

=> The "up-ramp" could be shortened, that is correct. However, if there is a need for a faster measurement than 240 Hz, we could easily increase the frequency. Our group has demonstrated measurements in the several 10 kHz range. (Witzel, 2013)

• **Line 501-506: Why does WMS specifically require lower fringe levels? This requirement would involve many factors, and the authors do not present any evidence or reference for the claim.**

=> WMS allows an effective filtering for noise in the raw signal. WMS behaves similarly to a bandpass filter. WMS does not directly require lower fringe levels. However, many WMS instruments only become more precise by using the lower noise advantage. The higher precision WMS achieves means that WMS instruments operate at much lower fractional optical densities often well below 10-4 down to 10-7 OD. Operating at such small optical densities automatically requires that the optical baseline of the instrument is improved accordingly. Thus WMS instruments will depend on the fringe levels to be much better suppressed than a less precise dTDLAS instrument. This demand for better fringe suppression brings a higher susceptibility for base line variations which leads to enhanced drift and systematic variations and the need for more frequent calibrations, which is exactly what we wanted to avoid. From our experience, dependence on a super stable baseline affects the absolute accuracy of an instrument. Such a high baseline stability is almost impossible to be ensured. Absolute accuracy stability (=long-term behavior) is a prerequisite for a calibration-free instrument; hence, we deliberately avoided operating HAI below fractional absorption resolutions of 10 -4 OD . In this context, a typical WMS instrument, operating in the E-5 to E-6 range, is typically much more affected by a E-5 fringe than a dTDLAS instrument operating in the E-3 to E-4 range. => We added some words to clarify

• **Line 509: ramp -> ram?**
=> corrected

• **Line 511-512: OP 6x greater 1s residual that CP? This doesn't make sense with**

==> Indeed, this statement is correct. Besides the arguments given in lines 511 – 513, one should regard chapter 5.2 about precision (submitted version). In short: The residual of the closed-path cell is mainly dominated by surface reflections leading to optical interference. The close-path cells are by their design mechanically very stiff. This leads to the large difference between the precision value determined via the "residual method" and the Allan method. Contrarily, the major contributions to the noise pattern of the open-path cell (Figure 8 submitted version) are wind/pressure turbulences, sun radiation, vibration, mechanical bending of open-path cell, dirt layer on mirror, scratches on mirrors. The greater 1s residual is therefore expected by us. => we added some words to clarify

**the Sect. 5.2 results**

• *Line 512-513: How does the wind speed affect this? Is it pressure deviation?*

=> This is linked to the question above: The "active" measurement cycle of HAI is approx. 1.4 ms, which corresponds at 800 km/h to 30 cm. From this point of view, HAI averages spatially water vapor values over theses 30 cm. If we had large influences of wind or pressure variations, we would see that in the raw data => Figure 9 (submitted version). Keep in mind, that we recalculate the Voigt-Profile. We would see if e.g. the gas pressure would not fit to the expected line broadening. Vice versa, the gas pressure sensor can be checked by line broadening. See our publication on optical pressure determination using HAI (Buchholz et al., 2014b) , where we discussed influences and the validation of the pressure sensor in detail.

**Cavitation causing misalignment?**
=> See figure 3 (submitted version): To be compatible with bird impact security requirements, the open-path cell was not allowed to be mechanically as stiff as we initially wanted. We had to cope with some flexing of

the open-path mirror holders, leading to small amounts of misalignment. This misalignment can be easily detected by reviewing raw data to determine the strength of the gross light transmission over the open-path. A critically important point we have to make and emphasize more in this paper is that HAI delivers and stores not only raw spectra but also more than 120 internal instrument's "housekeeping data." This allows for a very detailed analysis of the instruments' behavior. Therefore, HAI delivers much more than a single final water vapor concentration. HAI captures a very high temporal resolution and an entire status of the instrument, which is then used later to detect failures or problems with high efficiency and flexibility.

**Wall effects from the airfoils? Have you determined this from performing speed experiments on the aircraft?**
=> HAI is mounted in the fuselage and therefore far away from the airfoils. The speed envelope of an aircraft such as HALO is relatively narrow for each flight level. Speed variations from 90 km/h to 900 km/h in lower stratospheric regions are not possible since water vapor is homogenous enough to determine at least some kind of Allan variance. Usually, we don't see any correlations between flight speed / angle of attack and HAI's measurements. In situations where this happens, we can explain them: See e.g. (Buchholz et al., 2014b) where we could link gas pressure drops of 30% to a self-shading effect.

**• Line 513-520: This seems an odd place for this discussion. Also, much of the temperature and pressure difficulty with open path systems arises from the uncertainties in the temperature and pressure fields themselves, not instrumental artifacts, which would result in the same issue here. Spectral effect in temperature and pressure can be minimized through spectral line choice, which would also be similar for either method.**
=> We partly agree to this statement. WMS systems working in large pressure or concentration ranges use (at least the one we know in detail) a multi-dimensional calibration against pressure and temperature or correct the peak to trough vales via a simulated (c,p,t) matrix. Cal-free dTDLAS need gas pressure and temperature as measurements in the first place too. For temperature, we agree that the statement is true – at least for an instrument like HAI which does not measure the temperature optically (e.g. two line ratio with different line strength temperature dependencies) which we do for example in our earlier spectrometers for combustion applications. However, we demonstrated in (Buchholz et al., 2014b) , that we can validate internal pressure sensor in the open-path cell pretty well. The uncertainties calculations in this paper include this fact. To our knowledge, no open-path WMS sensor has shown such a pressure validation so far, and we doubt that it is possible on that accuracy level. For the internal close-path cell, the situation is entirely different. Due to the mechanical design (e.g. heat exchanger), we have (within the boundary of our uncertainties) a homogeneous temperature and pressure field. HAI's evaluation approach uses the actual temperature and gas pressure for evaluation. We revised the wording to make that entire statement clearer

**• Line 543-546: This precision is for 1.4 µm or 2.6 µm system? What is the precision of the other? Maybe a table to summarize which precisions/detection limits correspond to which channel**
=> Thank you – that was missing => table added

**• Line 598-601: it would be nice to see a figure with a greater summary of the intercomparisons between the 4 channels than the singular example shown in figure 10 + Line 678-684: this would benefit from an additional figure**

=> We deliberately avoided including a merged plot of many flights. We are aware that there are many papers like that merging data together to retrieve statements such as "5% averaged deviation over 200hours". However, we believe that a statement like that is dangerous and doesn't bring any benefit. HAI's philosophy is to decorate every single measured data point with an uncertainty including any impact at any time. We will focus on that in an independent, metrological based publication considering any impact independently for a closed-path cell and open-path cells, in-flight and in lab, as well as in comparison with other instruments. However, an analysis with such details would not fit in this paper which describes mainly the state of the art, and the setup of HAI. We feel that we first need to describe the concept, the properties and the peculiarities of HAI in this paper and provide sufficient information on validation and instrument behavior, before we should start to "average" and compress data over hundreds of hours. In our Experience using the very high data acquisition speed more of HAI there are plenty of situations, where the spatial variations in atmospheric water vapor are so strong that one has to look quite careful on the data to decide if spatial averaging is permitted or if it causes nonlinear averaging behavior. Until this is done we feel that it is much more elusive to look at shorter time frames and to discuss in case studies the problems and capabilities of HAI

**• Line 604-609: this statement should be made with the caveat that the 20**

Sorry, we didn't understand the question

**• Line 637-652**
**– The instrument in (c) has flown many campaigns on the NASA ER-2 (see JPL Laser Hygrometer, or JLH).**

=> Thanks for this hint

**– The authors omit the NASA DLH hygrometer (Diskin et al., 2002), which has frequently flown missions for over 20 years.**

=> We did not mention the paper from Diskin, since we could not find any "clear" statements about the accuracy or precision. The paper is a little bit unclear in its statements. e.g.: The calibration is done at 3 m. We assume at a 10 higher concentration than the target concentration (28.5/3). Measurements are shown up to 1000 ppmv. How is this calibration then corrected for e.g. self-broadening which is negligible at 10 ppmv but not at 10 000 ppmv?

Further, the publication (Podolske et al., 2003) mentioned: "An error analysis for DLH during this flight period, combining uncertainties in line strength (2.8%), 2f offset correction (2.3%), pressure (0.4%), and temperature (0.6%) gives a 1s error estimate of 3.7%. A conservative 2s error bar for the DLH data shown in Figure 6 would be ±0.75 ppmv, which spans all of the JPL ER-2 and Harvard ER-2 data during the first portion of the intercomparison. All this uncertainties seems from a metrological point of view challenging … even in a laboratory environment. Nevertheless, we now added these papers. (Diskin et al., 2002) and (Podolske et al., 2003).

**– Line 644-645: again recommend that aircraft be referred by their affiliation, not nationality. HAIPER -> HIAPER.**

=> changed

• **Figure 1: I'm not sure I see the need for this figure, other than to show that the same laser can be used for multiple channels simultaneously. I recommend removing it. Maybe the intent of this figure could be added to Fig. 4 by adding the parasitic water and stabilization channels.**
=> Figure 1 illustrates the fundamental setup of a multichannel TDLAS system and indicates which components can be put together (DAQ). It also includes which ones have to be included for each of the channels (Laser driver , detectors amplifiers). We think it is important to illustrate which and how many instrument components are integrated in the cabin part of the instrument. We do not see an advantage from removing this figure.

• **Figure 2: This would be more beneficial if this were split into two bottom axes, then zoomed in on each water line, showing only the lines immediately around it.**
This figure is essential to motivate the line selection of HAI with respect to interferent lines and species. We would therefore like to keep it. The information which the reviewer would like to see can be deduced from Figure 6 were the low pressure scans and from Figure 8. Both show that no strong neighboring lines are showing up. Thus we don't see a need to modify Figure 2. Instead we included a link to Figure 6 and figure 8

• **Figure 5: I don't feel this figure adds significantly to the manuscript, and I recommend removing it.**
It is important for the reader to understand the deliberate redundancies realized by the 2 by 2 channel approach. This approach is unique, has never been realized or incorporated in an airborne hygrometer before and enables the important cross validation possibilities of the HAI spectrometer. We think this figure is essential. We find that the paper would lose in clarity if we would remove figures as recommended by the reviewer. As AMT does not require a certain maximum manuscript length to be fulfilled we also don't understand the need to remove valuable information

• **Figure 7: cursing -> cruising**
=> changed

**Anonymous Referee #2**

This paper discusses an aircraft tunable laser spectrometer titled Hygrometer for Atmospheric Investigations (HAI) for in situ water vapor and total water measurements. The paper discusses the four separate measurements that HAI provides via two independent diode lasers (1.4 and 2.6 μm) and two different analysis cells (open and closed path). The authors discuss how HAI will improve upon present field instruments and show some flight data.

**Major Issues:**
• **The purpose of this manuscript is unclear. The present paper has 'multi-phase' in its title but does not discuss HAI total water components in sufficient detail. Such details are necessary to understand how the effectiveness of HAI multi-phase measurements. Total water measurements present new challenges such as particle enhancement factors and memory effects that require careful study – and none of that is presented in this manuscript. Rather, this paper is about how the methodology of multi-channel water measurements provide a more accurate water vapor measurement than prior art. In this light, there is**

**clearer justification for this manuscript – i.e. that measuring water vapor simultaneously via two different wavelengths and two different analysis cells helps constrain measurement uncertainty. However, the authors need to re-write the body of the manuscript to discuss this in more detail. For instance, the Harvard HHH and total water instrument provide a multi-channel measurement but that is not discussed in this manuscript. Also, past aircraft measurements involved multiple instruments on one aircraft – how does HAI do better than these experiments?**

=> Thank you very much for your comment. We will split your statement in the different topics to clarify it for an external reader:

a) *"multi-phase" vs. "multi-channel"*
   We use the following, commonly used definition:
   **Multi-channel** means that the same parameter is measured with/in different channels. E.g. HAI measures in both close-path cells the water vapor + evaporated ice content. Both cells are parallel to each other. (This is similar to the cited HHH instrument (Sargent et al., 2013a), which measures gas-phase with a Lyman-a and a TDL channel)
   **Multi-phase** means that we can detect different phases of water vapor such as gas phase and ice/droplet phase (from the difference between total and gas phase water).
   => See in submitted paper the entire chapter about the setup of HAI. In short: HAI measures with two independent spectrometers (1.4 µm and 2.6 µm) at two different locations (open-path, close-path). All spectrometer use the same technique: dTDLAS. Both, the 1.4 µm and 2.6 µm open- and closed-path measurements are each coupled via the SAME laser. This allows not just transferring the validation of the close-path cell to the open-path cell, it also allows a supervision of the laser-behavior - which would be difficult for an independent closed-path cell laser.

b) *total water measurements, particle enhancement factors*
   HAI (see concept description in submitted paper) measures total water (= defined by water wapor + ice/droplets). We didn't discuss particle enhancement factors because we assumed that inlet characterization papers such as (Krämer and Afchine, 2004) are known. Sampling issues are similar for every extractive sampling instrument, we assumed that it is commonly known that total water vapor measurements has to be corrected for that (as e.g. described in (Krämer and Afchine, 2004))
   => we will add a paragraph to clarify that.
   As a remark: Sampling analysis/characterization of different inlets is a very interesting topic and we entirely agree that the entire airborne water vapor community hasn't put enough attention to that topic in the past. HAI will (in a different configuration in a future campaign) be capable of analyzing sampling artefact when connected to different kind of inlet systems. We are looking forward to describe this in an upcoming paper after the campaign.

c) *memory effects*
   We agree (see line 123-131), that is a big issue - especially for close-path systems with low gas flows such as cavity ring down or mass spectrometer. HAI is deliberately optimized for a quite high gas flow. A typical value is (see line 131) for HAI is 100 (!) vol-liter/min. In addition (see line 130) HAI has 80 °C heated inlet lines. The inlet pipes are electro-polished stainless steel. The gas current in the internal measurement cell (successor of: (Kühnreich et al., 2016)) is perpendicular to the laser beam. This all together minimizes memory effects pretty efficiently. The open-path cell has no memory effect as the gas crosses the laser beam with a few hundred kilometers per hour. Therefore, by comparing open-path and closed-path cell, we can quantify memory effects in our inlet system and in the

cell on a very high level. We agree that, similar to a sampling analysis, memory effects are of high interest. Although, the paper focuses on the entire instrument and therefore it aims on the "entire" description and forces us not to describe every side issue/measurement advantage – even if that is impressive.

=> we added a few words about sampling.. (more on sampling effect quantification after the above mentioned campaign where HAI will be flown in a special non-multi-phase configuration )

d) _**Uncertainty: "that measuring water vapor simultaneously via two different wavelengths and two different analysis cells helps constrain measurement uncertainty."**_
We belief that the reviewer means the right thing: Measuring with an instrument like HAI guarantees the correctness (within the uncertainties) on a higher level compared to other instruments. However, the major contributions to the improved accuracy and robust evaluation are coming from the above mentioned "housekeeping data". These more than 120 independent measurement parameters allow at any time a clear assessment of the instruments status. A malfunction can be detected or much more important their absence can be proven. In the context of HAI, we use the metrological definition of "uncertainty" (JCGM 2008, 2008; Joint Committee for Guides in Metrology (JCGM), 2009).

e) _**"For instance, the Harvard HHH and total water instrument provide a multi-channel measurement but that is not discussed in this manuscript. Also, past aircraft measurements involved multiple instruments on one aircraft – how does HAI do better than these experiments?"**_
=> The Harvard HHH (Sargent et al., 2013a) (see image from paper) measures with TWO different principles (Lyman-a and TDLAS) the SAME value.

[Figure]

We agree, this is a fantastic instrument – especially for combining the benefits of Ly-a and TDLAS.

However, the first reviewer admonished that parts of the paper provide too much background. We try to find a balance - but simply not list every single instrument, and show all their benefits/drawbacks if they are not important for a comparison with HAI. The entire concept of HAI is significantly different to the one described in the HHH paper. Of course there are a lot of multi-phase measurements done in the past (see in paper 425 - 449) e.g. with two different WVSS-II (Abel et al., 2014) .

 To answer your question:      "how does HAI do better than these experiments"
=> this is described in line 80 to 230.

Asking this question in the context of a comparison between HHH and HAI? Generalized statements such as "Instrument X is better than Y" are not appropriate in a scientific context: E.g. the mentioned HHH covers a range of 3.5 to 600 ppmv 5% ± 0.7 ppmv (respectively 5% ± 0.2 ppmv). The 1.4μm path of HAI covers 3 to 40000 ppmv with a calculated uncertainty. Even these two statements cannot be compared: a) the range of HAI is larger => is HAI better? – No! HHH is designed specifically for LS and therefore a large water range is not needed. b) Uncertainty: In (Sargent et al., 2013b): "The uncertainty in the calibration of HHH is ±3% in slope based on the comparison with our laboratory standard outlined above, and the maximum potential offset is ±0.2 ppmv based on the agreement of HHH and Lyman-$\alpha$ in the lab. Including both calibration uncertainty and precision, we obtain a combined instrument uncertainty of 5% ± 0.2 ppmv (5% ± 0.7 ppmv during the MACPEX campaign)." A calibration with an uncertainty of 0.2ppmv is possible yes – but this is even for national metrology institutes specialized in metrological water vapor measurements still a huge (!) challenge. A prove that the described "laboratory standard" has the required feature and in particular accuracy and stability is missing. In particular it is unclear if the lab standard provides a similar degree of comparability than a national water vapor standard which HAS TO be validated via numerous comparisons on a global scale with other primary water standards before it is accepted as such. HAI instead has metrological calculated uncertainties. E.g. for offset uncertainty see entire procedure described in (Buchholz and Ebert, 2014b). We expect HAI to have a long-term stability <1% and an offset stability <1ppmv. Why is that not presented? In a paper which described the background, state of the art, setup, precision, signal analysis and flight data, there is just not enough room for a scientifically solid and valuable discussion of a metrological comparison. E.g. the determination of HAI's offset uncertainty is based on complex procedure (Buchholz and Ebert, 2014b) - HAI has this feature included. A "10-line description" would not be possible. This HAI paper cannot present all new features in the first paper.

A short but incomplete list of key words for other differences to HHH (please also read line 80 to 230 again): ): We use … the same technique for evaluation of open and closed-path, calibration-free - first principles - evaluation (NO water vapor calibration with respect to a reference), lower precision, long-term stable, self-validation, metrological uncertainty, faster time resolution, complete supervision of the instrument status, under-wing condition vs. air-conditioned cabin (which eliminated the risk of temperature impacts), total/gas-phase measurements with same principle, cross-checking capabilities between the different channels, offset determination of instrument, highly defined measurement conditions in close-path cells, etc.

=> We added on several positions additional information

**• "The term 'calibration-free' is misleading and the authors do not prove HAI is 'calibration-free'. At the percent or better level (accuracy and/or repeatability), even first-principles, direct absorption measurements need to consider instrumental artifacts such as detector non-linearity and low-pass filtering of the detector signal. Both of these artifacts can be misinterpreted as using the wrong line-shape algorithm. In this way, a first-principles methodology might be more prone to error than one that lumps all translation factors into one multi-parameter function.**

**It will take some work to prove a system to be 'calibration-free'. Such proof would be measuring a known concentration if non-stick gas (methane for the 1.4 μm channel) at several different concentrations, then repeating this for different temperatures the detector and filtering electronics will encounter."**

=> We appreciate that comment. Please read again the section (line 274 to 296) about the term "calibration-free". Please also read the Appl Phys paper on the side by side comparision of our dTDLAS approach with a primary water vapor generator (Buchholz et al., 2014a), to our knowledge the only metrological validation of the absolute accuracy of dTDLAS comparison for water vapor. This should clarify this misunderstanding entirely. Calibration-free does NOT mean "perfect down to the sub-percent level" … we never made this statement! We already wrote that in the text (line 293- 295):

"Everything has to be characterized in advance on such a level that the first measurement value is determined within its uncertainty limits." The evaluation concept (line 298 to 349) allows (line 546 - 608) an evaluation without any calibration **within the uncertainty** (!). The method was validated e.g. in the citied paper (Buchholz et al., 2014a). Deviations e.g. caused by line-shape deviations are also described in this paper. Detector nonlinearities have been found to be negligible for the used detectors (InGaAs, InGaAsExt). This might not hold for less ideal detectors like Mid IR MCT!

As a remark: Outside of metrology, calibration and validation mixed. (JCGM 2008, 2008) defines: "calibration (…) in a first step, establishes a relation between the measured values of a quantity with measurement uncertainties provided by measurement standards (…). HAI was never been calibrated by comparing the HAI instrument with any kind of reference water vapor source and comparing its output values to match the number given by the reference.

• **Lines 638-650 have incomplete information and need to be better researched and rewritten.**
**There are at least two other open path aircraft instruments not listed here: DLH (Disken et al.) and HHH (Harvard Anderson Group). Moreover, JLH (part c of author's list) has flown more than 200 times and there is a wealth of information on how it performed against other water instruments and satellite retrievals.**
=> Thank you - we added/changed (see comments above, first reviewer mentioned that too).
We did not mention the HHH here, because it is on the fringe of being a close-path system. We had to draw somewhere the line; our reference list is not meant to be complete … our introduction is already quite long.

**Minor issues:**

1) **The entire paper needs to be edited to be less conversational. For example, don't use phrases like "Of course" and re-write the sentence in lines 226-228 to be less conversational.**
 => we changed that, e.g. deleted the four "of course"
2) **Correct spelling of 'Herriott'**
=> corrected

3) **Define 't' in equation**
=> added

**Anonymous Referee #3**

**Buchholz and co-authors present a thorough and well written description of a thoughtfully-considered 2x2 spectroscopic measurement of water vapor and total water. I recommend publication after they address a few issues detailed below:**

**My major criticism concerns the total water channels. The inlet design parameters of 80 °C piping temperature and 100 liter/min flow rate (line 130) do not seem appropriate to ensure efficient conversion of condensed water to vapor prior to quantification in the closed-path cells. Any characterization of condensed phase sampling efficiency or comparison to other measurements during ESMVal would improve confidence in the accuracy of these measurements. If it is known that the total water channels have some inadequacies, this should be explicitly discussed at appropriate points within the manuscript, or alternatively, claims of multi-phase measurement capability should be tempered or removed.**

Thank you very much for your comment. Total water vapor measurements using a forward facing inlet system (such as HAI) are not genuinely new – instead this is the standard method on aircraft. Depending on the exact payload, the installation in HALO requires a total ½" tubing length of 3-5 m, with several curvatures and rectangle L-connectors, flexible elements etc. This facilitates a lot of surface to gas interactions in the piping. The final gas temperature (before the air enters the instrument) is approximately 5-10 K less than the heated tubing temperature. What's more, it has to be considered for heat capacity deliberations that 100 liter/min at 100 hPa is "just" 10 std liter / min. In addition, HAI internally splits this flow in two(!) parallel heat exchangers (shown in the picture below) to avoid any temperature inhomogeneity (for spectroscopic reasons).

[Figure]

In other words, compared to other typical airborne installing/ instrument setups, HAI can ensure much better a full evaporation of all ice/droplets than others. If one even doubt that with a setup like HAI, other well established setups such as (Zöger et al., 1999) cannot prove it either – this would basically mean one doubts pretty much all extractive total water measurements. If such problems occurred at a high level it w highly unlikely that the direct comparison of our open and closed channels yields such a good match after a cloud transect. If ice or liquid would not be evaporated completely it would quite likely build-up in the instrument and lead to long tails after a cloud transect, which we haven't found.

However, one point is very important, and we already added a paragraph to emphasize that. We didn't discuss "the sampling process," because we assumed that characterization papers such as (Krämer and Afchine, 2004) are known. Sampling issues are similar for every extractive sampling instrument, we assumed that it is commonly known that total water vapor measurements has to be corrected for that (as e.g. described in (Krämer and Afchine, 2004))

Sampling analysis/characterization of different inlets is a very interesting topic, and we entirely agree that the entire airborne water vapor community hasn't put enough attention to that topic in the past. HAI will (in a different configuration in a future campaign) be capable of analyzing sampling artifact when connected to different kind of inlet systems. We are looking forward to that. HAI is a powerful instrument with a very long list of novel features. This paper seeks to show an overview; more specialized future papers will describe in detail special features and special flight configurations. We are fully aware that we cannot show all features or all validations (metrological as well as comparisons to different communities) in this paper. Our philosophy is, and we believe that only in that way scientific instrument development knowledge is shared between scientists, that if we show data to prove a specific performance, it should explain everything in detail with all problems. However, this is not possible in a broader overview paper which has to fit to the page requirements of a Journal. E.g. we know from measurements that that the agreement of both total water measurements is better than 0.5% in flight - however explaining/proving this pretty tough statement would require an entire section with several validations. By doing that, the current HAI paper would "shift" from a "general description" to a displaying of one specific, single feature. Therefore, we strongly believe ( and hope for the understanding of the reviewers) that this paper has to convey the overview information, why e.g. a complex open-path pressure measurement such as (Buchholz et al., 2014b) is vital for accurate open-path hygrometry rather than a prove of concept study without further usage, and why e.g. we are so sure about HAI's open-path gas pressure readings. (Buchholz et al., 2014b) shows exemplary that complex features need their "holistic" descriptions – a 10 line description in this paper wouldn't be satisfying and raise more questions than answer.

**More minor improvements are suggested below.**
**In the section 'Explanation of the term "calibration-free", the authors are encouraged to add a discussion of the criteria they use to determine when and how frequently they re-characterize the component calibrations within the 'unbroken chain of [metrological] measurements' to assure the continued traceability of direct absorbance measurements.**
=> This is a good point - we added some information

**Reference lists of existing hygrometer publications exclude Zondlo [2010] in line 77 and in the reference cited in lines 94 and 645. Additionally on line 645, the American aircraft model is a Gulfstream-V (500).**
=> We added both + we emphasized that it is a non-exhaustive list.

**In line 650 and following, the May et al. developed hygrometer has flown on many NASA missions, supported by May and Webster as well as by Robert Herman. See, for example, data publication and instrument description found in A Hallar, et al., 2004 JGR-Atmospheres and DE Hagan, et al., 2004 GRL.**
=> Thank you very much - we have already revised that part of the paper.

**In the sentence beginning on line 661, would the .6%/K temperature influence on uncertainty**
**result in a +/- 3% effect on the open-path water vapor measurement (due to the 5K local temperature mesurement uncertainty)?**
=> Absolutely correct. Our assumption with 5K is conservative. With a full responsive (flight parameter) CFD model we could significantly reduce the uncertainty in the future. However, the static ambient temperature measurement accuracy, which is often assumed in the atmospheric community to be about 0.3 K, seems from a metrological point of view extremely (!) challenging to achieve. Gas temperature measurements on this level can be done in metrology environments for sure, but probably not as a total air temperature measurement at 900 km/h.

**In line 663, did the authors mean to write Voigt width, rather than Gaussian?**
=> The Voigt profile is a convolution of Gaussian and Lorentz profiles. The Gaussian broadening is affected by gas temperature change; the Lorentz broadening is affected by gas pressure and temperature.

**In the paragraph beginning on line 667, the systematic mean difference of the 2.6 closed-path: 1.4 closed path channels is more highly variable during the in-cloud transect and visual inspection would seem to imply that the mean difference undergoes a shift or temporal trend. An effort to filter both data sets to the same sampling intervals and frequencies prior to difference calculation would perhaps allow a more meaningful comparison between the mean channel difference during clear sky and in-cloud operations. It would not be surprising to see that the increased turbulence of in-cloud operations has differential impact on components of the 2x2 hygrometer, increasing the variability and perhaps even the mean of systematic channel differences.**
=> This is a very good point. The measurement data is already synchronized in time. The "noise" which is visible is caused by under-sampling. HAI's data in this plot is measured at 1 Hz. There is a lot more to say about the fine structure resolving capabilities of HAI, but that will need to be further reviewed in future papers. We tried to stay as close as possible to "standards" in the atmospheric community which are currently 1 Hz, standard sampling etc. The issue which is connected to that is a deep knowledge about the transfer behavior of the inlet system. Figure 10 gives a first impression of HAI's performance. A full analysis of the sampling system with statements about the high frequency sampling methods offered by HAI would shift the focus away from an overview paper. We will cover this in upcoming HAI papers.

**The authors claim in line 714 that the open-path and closed-path measurements show only a 2% difference in the same flight segment, however, Figure 10 shows larger differences, with instantaneous channel differences falling between +/- 3 to 5%.**
=> Thanks for the comment – a part of the sentence was missing (2% = average, 3-5% is deviation)

**Among the figures, it would be valuable to add a figure (after Fig. 8?) to show the fit quality of the open-path 2.6 um channel since it is the most likely channel to saturate.**
=> The strength of the absorption is shown in figure 6. Thus, we don't really see a need to add one more figure for this. Furthermore, we only use the 2.6 μm channel in pressure/concentration regions where the saturation is acceptable for the fit procedure. Scans which are excessively saturated lead to increasingly stronger systematic deviations and thus have to be filtered and suppressed. We investigated the possibility to evaluate strongly saturated spectra. However, this requires very high quality spectra (in the far Lorenzian wings) to maintain a calibration free evaluation. This was not the case during this flight campaign for the 2.6μm open-path channel due to low transmission conditions caused by imperfections of this novel type of SM-fiber. For significant parts of the campaign we therefore didn't use the 2.6μm open path, which was not too much of a problem due to the redundancy of the 1.4μm open path channel for most of the flight path.

We added a paragraph with a clearer statement.

**Finally, it would be useful to add a plot of relative channel deviation as a function of water vapor concentration over the same time interval as Figure 10 (13:40-13:55) to show that the channel differences are not related to calibration errors.**

=> This is a very good idea. We added a graph and a paragraph in the text.

As a small note: "not related to calibration errors" … since HAI is NOT calibrated at all, this would show instead if the relative deviation between the channels had a concentration dependence.